# The Role of the Vascular System in Degenerative Diseases: Mechanisms and Implications

**DOI:** 10.3390/ijms25042169

**Published:** 2024-02-11

**Authors:** Abdullah Md. Sheikh, Shozo Yano, Shatera Tabassum, Atsushi Nagai

**Affiliations:** 1Department of Laboratory Medicine, Faculty of Medicine, Shimane University, 89-1 Enya Cho, Izumo 693-8501, Japan; syano@med.shimane-u.ac.jp (S.Y.); tabassum@med.shimane-u.ac.jp (S.T.); anagai@med.shimane-u.ac.jp (A.N.); 2Department of Neurology, Faculty of Medicine, Shimane University, 89-1 Enya Cho, Izumo 693-8501, Japan

**Keywords:** degenerative diseases, vascular system, neurodegenerative disease, chronic kidney disease, cardiovascular disease, endothelial dysfunction, inflammation, age-related vascular changes

## Abstract

Degenerative diseases, encompassing a wide range of conditions affecting various organ systems, pose significant challenges to global healthcare systems. This comprehensive review explores the intricate interplay between the vascular system and degenerative diseases, shedding light on the underlying mechanisms and profound implications for disease progression and management. The pivotal role of the vascular system in maintaining tissue homeostasis is highlighted, as it serves as the conduit for oxygen, nutrients, and immune cells to vital organs and tissues. Due to the vital role of the vascular system in maintaining homeostasis, its dysfunction, characterized by impaired blood flow, endothelial dysfunction, and vascular inflammation, emerges as a common denominator of degenerative diseases across multiple systems. In the nervous system, we explored the influence of vascular factors on neurodegenerative diseases such as Alzheimer’s and Parkinson’s, emphasizing the critical role of cerebral blood flow regulation and the blood–brain barrier. Within the kidney system, the intricate relationship between vascular health and chronic kidney disease is scrutinized, unraveling the mechanisms by which hypertension and other vascular factors contribute to renal dysfunction. Throughout this review, we emphasize the clinical significance of understanding vascular involvement in degenerative diseases and potential therapeutic interventions targeting vascular health, highlighting emerging treatments and prevention strategies. In conclusion, a profound appreciation of the role of the vascular system in degenerative diseases is essential for advancing our understanding of degenerative disease pathogenesis and developing innovative approaches for prevention and treatment. This review provides a comprehensive foundation for researchers, clinicians, and policymakers seeking to address the intricate relationship between vascular health and degenerative diseases in pursuit of improved patient outcomes and enhanced public health.

## 1. Introduction

Systemic degenerative diseases, characterized by the gradual deterioration of tissues and organs over time, exert a significant influence on health and impose a substantial burden not only on individuals but also on society and healthcare systems. The impact of degenerative diseases is expanding due to the increased prevalence of aging populations. Conditions such as Alzheimer’s, Parkinson’s, cardiovascular diseases, and chronic kidney disease (CKD) become more common as life expectancy rises [1,2,3]. This demographic shift places immense pressure on healthcare systems and necessitates strategies to effectively manage and treat these chronic diseases.

Due to the impacts on the healthcare system and society, research efforts to comprehend the underlying mechanisms of degenerative diseases, develop effective treatments, and explore preventive measures are gaining importance. Such research highlights the importance of the interaction and interdependence of different systems on the pathology of systemic diseases. For communication between the systems, the body relies mainly on systems like the vascular system [4]. The vascular system plays a crucial role in multiple physiological processes throughout the body [4]. Dysfunctional vascular mechanisms often contribute to the initiation and exacerbation of diseases [5,6,7], highlighting the intricate relationship between vascular health and overall well-being. Disturbances in the vascular system can manifest as diseases of specific systems, such as the nervous system or kidney system [5,6,7]. Conversely, conditions of a specific system can reciprocally affect the vascular system [8,9]. Research into the emerging role of the vascular system in contributing to the development and progression of various degenerative diseases has gained significant attention in recent years. Here, we provide an overview of some key areas of research and findings in this field:Neurodegenerative Diseases: There is growing evidence linking vascular health to neurodegenerative diseases such as Alzheimer’s disease and Parkinson’s disease [10]. Research suggests that compromised blood flow to the brain, often due to conditions like hypertension and atherosclerosis, can contribute to cognitive decline and neurodegeneration.Cardiovascular Diseases and Metabolic Syndrome: The relationship between cardiovascular diseases and metabolic syndrome (a cluster of conditions like obesity, high blood pressure, high blood sugar, and abnormal cholesterol levels) is well-established [11]. These conditions are often intertwined and can collectively contribute to the progression of degenerative diseases.Osteoarthritis: Emerging research has started to explore the link between vascular health and osteoarthritis, a degenerative joint disease [12,13]. Poor blood supply to joint tissues may contribute to cartilage degeneration and joint inflammation.Age-Related Macular Degeneration (AMD): AMD is a leading cause of vision loss in the elderly. Studies have revealed associations between vascular factors, such as hypertension and atherosclerosis, and an increased risk of AMD [14].Chronic Kidney Disease (CKD): Vascular impairment plays a crucial role in the development and progression of CKD. Kidneys rely on a rich blood supply, and vascular damage can lead to renal dysfunction [7,15].Aging and Vascular Dysfunction: As people age, their blood vessels can undergo structural and functional changes, which can contribute to the development of various degenerative conditions [16]. Understanding the mechanisms behind age-related vascular dysfunction is a key area of research.Inflammation and Endothelial Dysfunction: Endothelial cells lining blood vessels play a crucial role in regulating vascular health. Dysfunction of these cells can lead to chronic inflammation and contribute to the development of degenerative diseases [17].

In this review, we discussed the detailed role of the vascular system in systemic degenerative diseases, with a primary focus on diseases of the cardiovascular, kidney, and central nervous systems (CNS), as they are integral in understanding the broader impact of vascular health on systemic degeneration.

## 2. Vascular System: Anatomy and Functions

### 2.1. Anatomy

The vascular system, also known as the circulatory system, is a complex network of blood vessels that circulate blood throughout the body, ensuring the delivery of oxygen, nutrients, hormones, and immune cells to tissues while removing waste products [4]. Comprising arteries, veins, and capillaries, the vascular system plays a pivotal role in maintaining physiological equilibrium [4].

Arteries: Arteries are thick-walled blood vessels that carry oxygenated blood away from the heart and distribute it to various tissues. They have a strong muscular layer that allows them to withstand the force generated by the heart’s contractions. The largest artery, the aorta, emerges directly from the heart and branches into smaller arteries that further divide into arterioles [18,19].Veins: Veins are blood vessels responsible for transporting deoxygenated blood back to the heart for oxygenation. Unlike arteries, veins have thinner walls and less muscular tissue. They use one-way valves to prevent blood from flowing backward and rely on skeletal muscle contractions to assist in pushing blood against gravity, particularly in the limbs. Veins gradually merge into larger vessels, ultimately forming the superior and inferior vena cava, which return blood to the heart [18,19].Capillaries: Capillaries are the smallest and most numerous blood vessels, connecting arteries and veins within tissues. They facilitate the exchange of gases, nutrients, and waste products between the blood and surrounding cells. Capillary walls consist of a single layer of endothelial cells, allowing for efficient diffusion of substances. Importantly, there are structural differences in the capillaries of different organs. For example, capillaries of the CNS possess tight-junctions, which allows them to be highly selective to the molecules that can enter CNS parenchyma. Oxygen and nutrients pass from capillaries into tissues, while waste products enter the capillaries for eventual elimination [18,19].

### 2.2. Functions

The vascular system serves a multitude of crucial functions, including (i) transporting oxygen, nutrients, and waste products, as well as (ii) maintaining homeostasis, which is fundamental to maintaining the overall health and function of the body [4,18,19] (Figure 1A). The relationship between the vascular system and degenerative diseases underscores the significance of its proper function in preventing and managing these conditions.

(i)Transporting Oxygen, Nutrients, and Waste Products: One of the primary functions of the vascular system is to facilitate the transportation of oxygen and nutrients to the tissues and organs. In the tissue capillaries, oxygen and nutrients diffuse from the blood into the surrounding cells, providing energy for cellular processes. Conversely, waste products, including carbon dioxide and metabolic byproducts, are exchanged at the capillary bed and flow back to the heart, where they are then pumped to the lungs for oxygenation and to other elimination organs for waste removal.(ii)Maintaining Homeostasis: The vascular system plays a crucial role in maintaining homeostasis through the regulation of body temperature by redistributing blood flow to dissipate or conserve heat. It also contributes to fluid balance by controlling the movement of water and electrolytes between blood and tissues. In response to injuries, the vascular system initiates clotting processes to prevent excessive bleeding. Additionally, blood vessels participate in immune responses by transporting immune cells to areas of infection or injury, contributing to the body’s defense mechanisms.

## 3. Vascular Dysfunction in Degenerative Diseases

Vascular dysfunction encompasses a range of impaired functions within blood vessels, including endothelial dysfunction, inflammation, oxidative stress, impaired angiogenesis, and irregular vessel structure [20,21,22]. In the context of degenerative diseases, vascular dysfunction assumes profound significance as it often serves as a common element connecting various disease processes across different body systems (Figure 1B). The compromised blood flow, insufficient nutrient delivery, and impaired waste removal linked to vascular dysfunction contribute to tissue damage, cellular stress, and declining functionality (Figure 1B). For instance, in cardiovascular diseases, vascular dysfunction plays a pivotal role in the development of conditions like atherosclerosis, hypertension, and heart failure [23,24,25]. In neurodegenerative diseases, such as Alzheimer’s disease, vascular dysfunction disrupts the delicate balance of nutrient supply and waste removal, including the clearance of amyloid β (Aβ) in the brain [26,27]. This acceleration of neurodegeneration contributes to cognitive decline. Moreover, vascular dysfunction significantly influences renal degenerative diseases. In conditions like chronic kidney disease (CKD) and diabetic nephropathy, compromised blood flow within the kidneys leads to tissue damage and reduced filtration capacity [28,29]. Endothelial dysfunction and inflammation further exacerbate kidney dysfunction, underscoring the critical role of the vascular system in maintaining renal health. Recognizing the role of vascular dysfunction has profound implications for understanding the pathologies of degenerative diseases. In this section, we discussed the specific contributions of vascular dysfunction to degenerative diseases within different body systems.

### 3.1. Cardiovascular System

Impaired vascular function plays a pivotal role in the onset and progression of cardiovascular degenerative diseases, including hypertension, atherosclerosis, and heart failure. These conditions, collectively known as cardiovascular diseases (CVD), impose a significant burden on healthcare systems in terms of morbidity, mortality, and associated costs. An estimate suggests that globally, CVD accounted for 19.05 million deaths in 2020, marking an 18.71% increase from 2010. The prevalence of CVD cases in 2020 was 607.64 million, indicating a rise of 29.01% from 2010 [30]. In the subsequent year (2021), the estimation of CVD-related deaths increased to 20.5 million, representing close to one-third of all global deaths, showcasing an upward trend that began during the 1990s [31]. The prevalence of CVD also showed an increasing trend, reaching 621 million in 2021, up by 2.19% from 2020 [32].

CVD not only exacts a social toll through disease occurrence and mortality but also places a significant strain on the economy and healthcare systems. In the USA, the cost of cardiovascular diseases amounted to USD 555 billion in 2015, comprising USD 318 billion in direct costs and $237 billion in indirect costs [33]. This figure is projected to escalate to USD 1.1 trillion by 2035. Moreover, the costs associated with informal caregiving for CVD patients were estimated at USD 61 billion in 2015 and are expected to rise to USD 128 billion by 2035 [33]. Notably, approximately four out of every five CVD-related deaths occur in low- and middle-income countries, exacerbating the strain on already overstretched healthcare systems in those regions [31]. Consequently, the economic burden of CVD could significantly impede healthcare systems, impacting not only CVD management but also other health conditions. To mitigate such immense strain on the healthcare system and the economy, preventive measures to improve vascular health could be crucial in managing the burden of CVD in both sectors.

To improve vascular health, a thorough understanding of vascular functions and how their dysfunctions contribute to the onset of CVD is essential. The intricate relationship between vascular dysfunction and conditions such as hypertension, atherosclerosis, and heart failure, which are described below, underscores the importance of maintaining healthy blood vessels for the overall management of CVD.

(i)Hypertension: Vascular dysfunction is a key contributor to the onset and maintenance of hypertension. Endothelial dysfunction is frequently seen in hypertensive conditions [24], where it results in narrowed blood vessels and elevated systemic resistance, causing blood pressure to increase [30]. Additionally, vascular dysfunction and the renin–angiotensin–aldosterone system, a hormonal pathway that regulates blood pressure, can affect each other bidirectionally. Dysregulation of this system due to impaired vascular function can lead to sustained high blood pressure levels, further damaging blood vessel walls and promoting hypertensive complications [31,32].(ii)Atherosclerosis: Impaired vascular function is intimately linked to the development of atherosclerosis, a condition characterized by focal inflammation and the accumulation of oxidized-LDL-laden plaques within arterial walls [33]. Endothelial dysfunction triggers inflammatory processes, attracting immune cells, including macrophages and T-cells, that contribute to plaque formation [23,34]. As plaques grow, the arterial lumen narrows, reducing blood flow and causing turbulence that further enhances the endothelial inflammatory condition. This cycle of inflammation, plaque buildup, and arterial narrowing creates a hostile environment conducive to blood clot formation, increasing the risk of myocardial infarction and strokes.(iii)Heart Failure: Vascular dysfunction significantly contributes to the progression of heart failure [25]. Chronic hypertension and atherosclerosis, driven by impaired vascular function, strain the heart by requiring it to pump against increased resistance, leading to cardiac enlargement, especially left ventricular hypertrophy. Additionally, vascular dysfunctions impact the blood vessels within the heart, which can lead to reduced oxygen and nutrient delivery, further weakening cardiac function [35]. The increased workload, impaired coronary blood flow, and inflammatory signals can also contribute to the development of heart failure with preserved ejection fraction (HFpEF), a type of heart failure characterized by impaired relaxation of the heart’s chambers [36].

### 3.2. Nervous System

Impaired vascular function plays a significant role in the development and progression of degenerative diseases affecting the brain, including Alzheimer’s disease, stroke, cerebral small vessel diseases, and Parkinson’s disease. The intricate interplay between vascular health and these neurological conditions highlights the importance of maintaining optimal blood vessel function for brain well-being.

(i)Alzheimer’s Disease: Vascular dysfunction is closely linked to the pathogenesis of Alzheimer’s disease (AD), a neurodegenerative disorder characterized by cognitive decline and memory loss. Deposition of aggregated amyloid β (Aβ) peptide is considered the main cause of AD [37]. In normal brains, Aβ is cleared mainly through perivascular pathways [38]. Dysfunctional blood vessels contribute to reduced Aβ clearance, leading to increased aggregation and deposition [5]. Such deposited Aβ further impairs cerebral blood flow and causes pathological angiogenesis, blood–brain barrier disruption, neuroinflammation, and impaired clearance of toxic protein aggregates, including aggregated Aβ peptide [5,6,39].(ii)Stroke: Impaired vascular function significantly increases the risk of stroke. Endothelial dysfunction compromises blood vessel dilation and responsiveness, promoting the development of blood clots [40]. Atherosclerosis, driven by vascular dysfunction, can lead to plaque rupture, triggering clot formation and embolic strokes [33]. Furthermore, chronic hypertension could cause sclerosis and damage the blood vessels, increasing the likelihood of vessel rupture and hemorrhagic strokes [41]. The role of vascular dysfunction in both ischemic and hemorrhagic stroke underscores its impact on overall brain health.(iii)Cerebral Small Vessel Diseases: Cerebral small vessel diseases encompass a group of conditions that affect the small blood vessels within the brain. Impaired vascular function contributes to the development of these diseases, including arteriolosclerosis and cerebral microbleeds [42]. Endothelial dysfunction, inflammation, and oxidative stress compromise the structural integrity of these small vessels, leading to vessel wall thickening, narrowing, and increased fragility [42,43,44]. The cumulative effect of vascular dysfunction in cerebral small vessel diseases disrupts cerebral blood flow, which can result in lacunar infarcts and contribute to cognitive impairment, mobility issues, and other neurological deficits.(iv)Parkinson’s Disease: Vascular dysfunction also holds implications for Parkinson’s disease (PD), a neurodegenerative disorder characterized by motor symptoms like tremors and rigidity. In PD, the primary pathology involves the degeneration of dopaminergic neurons in the substantia nigra region of the midbrain. The key diagnostic histopathological features of PD are intraneuronal Lewy bodies containing aggregated α-synuclein, leading to the belief that such aggregation is the primary cause of neurodegeneration and, consequently, the disease [45]. In the context of aggregation, an altered protein quality control system, especially the ubiquitin-proteasome pathway, is thought to play a crucial role in PD [46,47]. Numerous studies have implicated vascular changes, including impaired blood-brain barrier function and pathological angiogenesis, in the development of the pathological changes seen in PD [48,49,50,51]. These changes result in hypoperfusion. Given that hypoperfusion can impact the protein quality control system, including the ubiquitin-proteasome system, it is plausible that decreased blood flow to the substantia nigra may contribute to α-synuclein aggregation. Moreover, hypoperfusion can subject cells to stress, potentially triggering the aggregation process of α-synuclein. Furthermore, impaired vascular health disrupts the supply of oxygen and nutrients, which can expedite neuronal damage in this vulnerable region.

### 3.3. Kidney System

Impaired vascular function plays a significant role in the development and progression of various kidney degenerative diseases, including chronic kidney disease (CKD), diabetic kidney disease, and hypertensive nephropathy. The intricate relationship between vascular health and these conditions underscores the importance of maintaining optimal blood vessel function for kidney well-being.

(i)Chronic Kidney Disease (CKD): Impaired vascular function contributes to the pathogenesis and progression of CKD. The disease is a long-term medical condition characterized by the gradual and irreversible deterioration of kidney function over an extended period, typically months to years. CKD is often categorized into different stages based on the level of kidney function, with stage 1 being the mildest and stage 5 representing end-stage renal disease (ESRD), where kidney function is severely impaired, and patients usually require dialysis or a kidney transplant to survive [52]. Common causes of CKD include diabetes, hypertension, glomerulonephritis, polycystic kidney disease, and certain other medical conditions [53]. Hypertension can lead to increased resistance to blood flow. This can damage the delicate blood vessels in the kidneys over time, impairing their ability to filter waste and regulate fluids [54]. Additionally, atherosclerosis and endothelial dysfunction are often seen in individuals with CKD, which affect the kidney perfusion and ability to regulate blood flow and maintain an appropriate balance of vasodilation and constriction [54,55]. This results in reduced blood flow to the kidneys. Moreover, chronic inflammation associated with vascular dysfunction can promote fibrosis and scarring within the kidney tissues. As a result, kidney function gradually declines over time.(ii)Diabetic Kidney Disease: Diabetic kidney disease, also referred to as diabetic nephropathy, represents a prevalent complication of diabetes. Poorly controlled diabetes can directly inflict harm upon the glomeruli and renal blood vessels [56]. Additionally, advanced glycation end products (AGEs) can harm blood vessels by triggering the generation of reactive oxygen species (ROS) and fostering inflammation [56,57]. Elevated blood sugar levels result in injury to the blood vessels that provide the kidneys, ultimately causing endothelial dysfunction and inflammation.(iii)Hypertensive Nephropathy: Hypertensive nephropathy results from chronic hypertension damaging the blood vessels within the kidneys [58]. Vascular dysfunction plays a key role in its progression. Elevated blood pressure causes structural changes in the small blood vessels, leading to arteriosclerosis and narrowing of the renal arteries [59]. These changes reduce blood flow to the kidneys, triggering a cascade of events that contribute to kidney damage. The decreased blood flow prompts the kidneys to release hormones that raise blood pressure further, creating a vicious cycle [58,59]. The compromised blood vessels also impair the kidneys’ ability to filter waste and maintain fluid and electrolyte balance, ultimately leading to kidney dysfunction.

From these findings, it is evident that impaired vascular function significantly contributes to the development and progression of cardiovascular, neurological, and kidney degenerative diseases. Furthermore, the vascular system exerts a similar impact on all other systems of the body. The consequences of endothelial dysfunction, inflammation, and reduced blood flow within the organs or tissues culminate in tissue damage, fibrosis, and functional decline. Recognizing the pivotal role of vascular health underscores the importance of preserving blood vessel function as a potential target for preventing and managing degenerative conditions. Strategies aimed at improving vascular health, which encompass blood pressure management, glycemic control, and lifestyle modifications, hold promise for mitigating degenerative disease progression and enhancing health outcomes.

## 4. Molecular Mechanisms Linking Vascular Dysfunction and Degenerative Diseases

The molecular mechanisms linking degenerative diseases and vascular dysfunction are complex and multifaceted, and they can vary depending on the specific disease and organ system involved. However, some common molecular mechanisms are involved in the interplay between vascular dysfunction and degenerative diseases (Figure 2).

### 4.1. Endothelial Dysfunction

The properties of healthy endothelial cells are anti-proliferative to smooth muscle cells, anti-coagulative, vasodilatory, and anti-adhesive to inflammatory cells [60,61]. Endothelial dysfunction is a complex process involving various molecular mechanisms, signaling pathways, and molecules [61]. It is a key contributor to the development and progression of many vascular diseases, including atherosclerosis, hypertension, and diabetic vascular complications [61,62,63]. Figure 3 depicted the common molecular mechanisms of endothelial dysfunction. Molecular mechanisms involved in endothelial dysfunction include:(i)Oxidative Stress: Excessive production of reactive oxygen species (ROS), such as the superoxide anion (O_2_^·−^) and hydrogen peroxide (H_2_O_2_), can inflict damage upon endothelial cells and impair the bioactivity of nitric oxide (NO) [64]. Nitric oxide is a molecule that fosters vasodilation, thereby promoting vascular health [64]. Also, superoxide can interact with NO and generate peroxynitrite (ONOO^−^), which can similarly inflict harm upon these cells [65]. The activation of NADPH oxidases (NOX) and xanthine oxidase is commonly implicated in the generation of these ROS species, a phenomenon observed within endothelial cells in numerous degenerative diseases [66,67].(ii)Inflammation: Chronic inflammation, characterized by elevated levels of proinflammatory cytokines (such as TNF-α, IL-1β), chemokines (MCP-1), adhesion molecules (VCAM-1, ICAM-1), and inflammatory receptors (Toll-like receptors, TNF receptors), can be induced in inflammatory conditions [17,21]. Through the action of adhesion molecules, chemokines, and their receptors, proinflammatory cells are recruited to the affected area. Inflammatory cytokines and their receptors typically activate proinflammatory transcription systems, including NF-κB, AP-1, and STATs [68]. This activation leads to the production of more proinflammatory cytokines, chemokines, and growth factors in these cells, as well as in endothelial cells, disrupting their normal function. Additionally, inflammatory mediators can impair the synthesis of nitric oxide (NO) and promote vasoconstriction [69]. These effects create a vicious cycle that profoundly alters endothelial cell properties. Furthermore, low-density lipoprotein (LDL) cholesterol can undergo oxidation by reactive oxygen species (ROS) and infiltrate the endothelium. This triggers an inflammatory response and the formation of atherosclerotic plaques, which can narrow and stiffen blood vessels [33].(iii)Nitric Oxide (NO) Bioavailability: Nitric oxide (NO), synthesized by endothelial nitric oxide synthase (eNOS), is a key endogenous vasodilator crucial for maintaining proper vascular function [70]. Dysregulation of eNOS activity, whether through mechanisms like eNOS uncoupling or reduced eNOS expression, results in reduced NO bioavailability [69,70,71]. This, in turn, hinders vasodilation and promotes vasoconstriction.(iv)Endoplasmic Reticulum Stress: Accumulating evidence suggests that endoplasmic reticulum (ER) stress is one of the primary causes of endothelial dysfunction [72]. The buildup of misfolded proteins within the ER triggers unfolded protein response (UPR) pathways, which in turn contribute to endothelial dysfunction, inflammation, and apoptosis [72].

### 4.2. Atherosclerosis

Atherosclerosis is a common feature of many degenerative diseases, particularly cardiovascular diseases. It involves the accumulation of lipids, including oxidized LDL, immune cells, and fibrous tissue, within the arterial walls, leading to narrowing of the lumen and hypoperfusion supplying area and affecting the degenerative process [23,33]. Additionally, the molecular mechanism of atherosclerosis is not limited to the cardiovascular system; it can have systemic effects and contribute to the pathogenesis of degenerative diseases in multiple organ systems [55]. Here are some of the key signaling pathways and mechanisms through which atherosclerotic vessels can impact other systems:(i)Inflammation and Immune Responses: Cytokines and chemokines: Atherosclerosis is characterized by chronic inflammation within arterial walls. Inflammatory mediators, such as cytokines (e.g., TNF-α, IL-1β) and chemokines, can enter the bloodstream, leading to systemic inflammation [33,73].(ii)Immune Cell Activation: In response to atherosclerotic plaques, immune cells like T-cells, monocytes, and macrophages become activated and release proinflammatory cytokines [74]. These activated immune cells can then circulate throughout the body, contributing to systemic inflammation [23,33,74].(iii)Endothelial Dysfunction and Impaired Nitric Oxide (NO) Bioavailability: Atherosclerosis can lead to endothelial dysfunction, reducing the production and availability of NO, a molecule important for vasodilation and maintaining vascular health [70,71]. This endothelial dysfunction can affect blood flow throughout the body.(iv)Oxidative Stress: Atherosclerotic vessels generate high levels of reactive oxygen species (ROS) through processes like LDL oxidation and activation of NADPH oxidases [63,64]. These ROS can cause oxidative stress, which can damage tissues and contribute to the development of systemic diseases.(v)Platelet Activation and Thrombosis: Atherosclerotic plaques can rupture, leading to the exposure of pro-thrombotic substances. Platelets can become activated and contribute to thrombosis, potentially causing strokes or myocardial infarctions [75].

The systemic effects of atherosclerotic vessels on degenerative diseases in other systems underscore the importance of a holistic approach to managing and preventing atherosclerosis. Addressing risk factors and promoting vascular health can have a significant impact on overall health and reduce the risk of degenerative diseases in multiple organ systems.

### 4.3. Hypertension

The molecular signaling pathways through which hypertension affects degenerative diseases are complex and can vary depending on the specific disease and tissues involved. However, several key molecular mechanisms and signaling pathways have been identified that contribute to the relationship between hypertension and degenerative diseases:(i)Renin–Angiotensin–Aldosterone System (RAAS): The RAAS is a central regulator of blood pressure and fluid balance [76]. In hypertension, there is often an overactivation of the RAAS. This system involves the release of renin from the kidneys, which leads to the conversion of angiotensinogen to angiotensin I and then angiotensin II. Angiotensin II is a potent vasoconstrictor and can promote inflammation and oxidative stress, contributing to vascular damage, atherosclerosis, and organ damage. Angiotensin II also stimulates the release of aldosterone, which can lead to sodium and water retention, further increasing blood pressure and strain on the cardiovascular system [76,77,78].(ii)Inflammation Signaling: Hypertension is associated with chronic low-grade inflammation [17,79,80]. Molecular signaling pathways involving cytokines, chemokines, and inflammatory mediators, such as IL-1β, NF-kB, and TNF-α, play a role in the inflammatory response [80,81]. This chronic inflammation can contribute to the development and progression of degenerative diseases by damaging tissues and promoting atherosclerosis.(iii)Oxidative Stress: Increased blood pressure can lead to increased production of reactive oxygen species (ROS) in blood vessels and tissues. Molecular pathways related to oxidative stress, such as those involving NADPH oxidase and superoxide dismutase, are implicated [82]. Oxidative stress can damage cellular components, including DNA, lipids, and proteins, contributing to cellular dysfunction and degenerative diseases.(iv)Endothelial Dysfunction: Hypertension can impair the function of endothelial cells that line blood vessels [24,30,32]. Signaling pathways involving nitric oxide (NO), which is a vasodilator, and endothelin-1 (ET-1), a vasoconstrictor, are altered in endothelial dysfunction. This dysfunction can lead to vasoconstriction, inflammation, and atherosclerosis.(v)Cellular Growth and Hypertrophy: Chronic high blood pressure can lead to hypertrophy of cardiac muscle cells through signaling pathways involving proteins like mTOR (mammalian target of rapamycin) and calcineurin [83]. Cardiac hypertrophy can eventually lead to heart failure and increase the risk of cardiac degenerative diseases.(vi)Neuroinflammation and Brain Damage: In the context of hypertension-related brain damage and neurodegenerative diseases, signaling pathways involving microglia activation, cytokines, and amyloid-beta protein processing are implicated [84]. Chronic hypertension can promote neuroinflammation and contribute to the pathogenesis of conditions like Alzheimer’s disease and vascular dementia [84,85].

### 4.4. Environment and Endocrine Disruptors

The environment in which we live exerts a profound influence on our health. Pollution, exposure to toxic chemicals, dietary habits, and lifestyle choices all contribute to the complex interplay between environmental factors and disease development [86]. Of particular concern are endocrine disruptors, substances that interfere with hormone function and have been implicated in a myriad of health problems. These disruptors, found in plastics, pesticides, personal care products, and other consumer goods, can disrupt the delicate balance of the endocrine system, potentially contributing to both vascular disease and degenerative conditions. Mounting evidence suggests that exposure to endocrine disruptors may increase the risk of vascular diseases such as atherosclerosis and hypertension [87]. Chemicals like bisphenol A (BPA), phthalates, and polychlorinated biphenyls (PCBs) have been linked to endothelial dysfunction, inflammation, and oxidative stress, all of which are key drivers of vascular pathology [87,88]. Moreover, endocrine disruptors can cause dysregulated vascular smooth muscle cell responses, further exacerbating the progression of vascular disease [89].

In addition to their impact on vascular health, endocrine disruptors have also been implicated in the pathogenesis of degenerative diseases. Epidemiological and experimental studies have revealed associations between exposure to these compounds and the development of Alzheimer’s, Parkinson’s, and certain types of cancer [90,91,92,93]. Disruption of hormone signaling pathways and insulin resistance, endothelial dysfunction, oxidative stress, chronic inflammation, alterations in lipid metabolism, and mitochondrial dysfunction are among the proposed mechanisms by which endocrine disruptors contribute to degenerative conditions [94,95,96]. Given the interconnected nature of vascular disease and degenerative diseases, it is plausible that common environmental factors, including endocrine disruptors, may underlie their shared pathophysiology.

## 5. Common Risk Factors That Affect Vascular System and Degenerative Diseases

Many common risk factors contribute to both vascular dysfunction and the development of degenerative diseases. These risk factors often overlap and can exacerbate one another, increasing the overall risk of various health problems. Here, we discussed some of the key risk factors that are associated with both vascular dysfunction and degenerative diseases:Aging: The aging process itself is a significant risk factor for both vascular dysfunction and degenerative diseases [97,98]. As people age, their blood vessels naturally undergo structural and functional changes, becoming less elastic and more susceptible to damage [99]. This contributes to conditions like arterial stiffness and an increased risk of atherosclerosis, which can affect both the heart and other organs.Hypertension: Hypertension is a common risk factor for both vascular dysfunction and degenerative diseases [24,30,41,43,54]. Chronic high blood pressure can damage the walls of arteries, making them less elastic and more prone to atherosclerosis. It also increases the risk of conditions like stroke, heart disease, and kidney disease.Smoking: Smoking is a major risk factor for vascular dysfunction and degenerative diseases [99,100]. It constricts blood vessels, increases oxidative stress, increases blood coagulability, reduces oxygen delivery to tissues, and promotes the development of atherosclerosis. Smoking is strongly associated with cardiovascular diseases, kidney diseases, lung diseases, and cancers.Dyslipidemia: An imbalance in blood lipid levels, particularly elevated levels of low-density lipoprotein (LDL) cholesterol and reduced levels of high-density lipoprotein (HDL) cholesterol, is also considered a risk factor for various health conditions, including vascular dysfunction, atherosclerosis, nervous system diseases such as cerebral small vessel diseases, and diabetes [33,101,102]. Dyslipidemia, characterized by these lipid abnormalities, poses a significant risk for atherosclerosis, a condition that can affect arteries throughout the body and contribute to vascular dysfunction, as well as degenerative diseases like heart disease and stroke. Additionally, dyslipidemia is known to increase overall levels of inflammation and oxidative stress, negatively impacting not only the health of blood vessels but also the well-being of other organs.Diabetes: Diabetes, especially type 2 diabetes, is a common risk factor for both vascular dysfunction and degenerative diseases [29,51,62,103]. High blood sugar levels can damage blood vessels (diabetic vasculopathy), contribute to atherosclerosis, and increase the risk of heart disease, stroke, and peripheral vascular disease, along with degenerative diseases of the nervous system, including Alzheimer’s disease.Obesity: Obesity is associated with several health issues, including insulin resistance, dyslipidemia, inflammation, and increased blood pressure [104]. These factors collectively contribute to the development of vascular dysfunction and degenerative diseases, such as heart disease, type 2 diabetes, and osteoarthritis.Physical Inactivity: A sedentary lifestyle is a risk factor for both vascular dysfunction and degenerative diseases [105]. Lack of physical activity can lead to weight gain, worsen insulin sensitivity and diabetes, and promote the development of atherosclerosis. Regular exercise, on the other hand, can help improve vascular health and reduce the risk of many degenerative diseases.Poor Diet: A diet high in saturated and trans fats, refined sugars, and processed foods can contribute to obesity, dyslipidemia, diabetes, and inflammation [106]. These dietary factors are associated with an increased risk of both vascular dysfunction and degenerative diseases.Chronic Inflammation: Chronic inflammation is a common factor underlying many degenerative diseases and can also contribute to vascular dysfunction [15,21,34]. Inflammatory processes can damage blood vessel walls, promote atherosclerosis, and increase the risk of conditions like rheumatoid arthritis and cardiovascular disease.Stress: Chronic stress can contribute to high blood pressure, unhealthy behaviors (e.g., overeating or smoking), and the release of stress hormones that can affect blood vessel function [106,107]. Over time, this can contribute to both vascular dysfunction and degenerative diseases.

## 6. The Impact of Degenerative Diseases on Vascular Dysfunction

The relationship between the vascular system and degenerative diseases is bidirectional and complex. Vascular dysfunction can contribute to the development and progression of degenerative diseases, while degenerative diseases can, in turn, worsen vascular health. Recognizing and addressing vascular risk factors and promoting vascular health through lifestyle measures are essential steps in preventing and managing both vascular dysfunction and degenerative conditions. Here, we discussed some of the degenerative diseases that impact vascular dysfunction:Diabetes: Diabetes, particularly type 2 diabetes, can lead to vascular dysfunction [62]. High blood sugar levels can damage the endothelial cells lining blood vessels, promoting atherosclerosis and increasing the risk of heart disease, stroke, and peripheral vascular disease.Inflammation: Chronic inflammation, a hallmark of many degenerative diseases (e.g., rheumatoid arthritis, inflammatory bowel disease), can directly impact the vascular system [108,109]. Inflammatory processes can damage the endothelium, leading to endothelial dysfunction and increased risk of atherosclerosis and cardiovascular events.Osteoarthritis: In this degenerative joint disease, the wear and tear on joints can lead to inflammation and damage to nearby blood vessels [12,13]. Chronic inflammation can contribute to vascular dysfunction in affected joints, exacerbating pain and limiting mobility.CKD: CKD is both a cause and consequence of vascular dysfunction [52,53,54,55]. Kidney damage can result from prolonged exposure to risk factors like high blood pressure and diabetes, which also contribute to vascular impairment. In turn, kidney dysfunction can lead to imbalances in electrolytes and hormones, affecting blood vessel tone and function.Cerebral Small Vessel Diseases: CSVD primarily affects the small blood vessels in the brain, leading to white matter lesions, lacunar infarcts, and microbleeds [42,43,44]. These vascular abnormalities are closely associated with cognitive decline, and they can disrupt the brain’s ability to function properly. CSVD can lead to both ischemic and hemorrhagic events in the brain. These events contribute to vascular dysfunction and can result in stroke-like symptoms and cognitive impairment. Degenerative changes in small blood vessels can lead to reduced blood flow, increasing the risk of strokes and vascular dementia.Alzheimer’s Disease: In AD, cerebral vessel integrity is compromised due to the deposition of Aβ, resulting in cerebral amyloid angiopathy [27,42]. These abnormal protein deposits can affect blood vessels, reducing cerebral blood flow. Reduced blood flow deprives brain cells of oxygen and nutrients, contributing to cognitive decline and memory loss. Some individuals with Alzheimer’s disease also experience vascular cognitive impairment (VCI), where vascular dysfunction in the brain exacerbates cognitive deficits. This can occur alongside the neurodegenerative processes in Alzheimer’s disease.

## 7. Therapeutic Implications

Targeting vascular dysfunction is a promising approach to mitigate the progression of degenerative diseases, including conditions like chronic kidney disease (CKD). Vascular dysfunction plays a critical role in the pathogenesis of many degenerative diseases, and addressing it can have a positive impact on disease management. Here, we discussed some potential therapeutic interventions for degenerative diseases aimed at targeting vascular dysfunction:Blood Pressure Control: Maintaining optimal blood pressure is crucial in managing vascular dysfunction. Antihypertensive medications, such as angiotensin-converting enzyme (ACE) inhibitors and angiotensin II receptor blockers (ARBs), are often used to control hypertension and reduce the strain on blood vessels [110]. They can also help control CKD [111]. Although antihypertensives are not the primary treatment for CSVD, associated hypertension may worsen the disease, and hypertension control can show beneficial effects. Some studies have suggested a link between hypertension and an increased risk of developing Alzheimer’s disease [112,113]. Hypertension may contribute to vascular dysfunction in the brain, which can exacerbate AD-related cognitive decline.RAAS Inhibition: Inhibition of the renin–angiotensin–aldosterone system (RAAS) is a common therapeutic strategy to protect vascular health in conditions like CKD. Medications like ACE inhibitors and ARBs not only lower blood pressure but also have direct vascular protective effects by reducing vasoconstriction and inflammation [111,114].Antioxidant Therapy: Since oxidative stress plays an important role in most degenerative diseases and vascular dysfunction, antioxidant therapy could be a good adjunct to the main therapy of a specific degenerative disease [64,66,67]. Antioxidant supplements or diets rich in antioxidants may help combat oxidative stress, which is a common contributor to vascular dysfunction. Antioxidants like vitamin C, vitamin E, and coenzyme Q10 may have beneficial effects in vascular diseases as well as neurodegenerative diseases, including Alzheimer’s disease.Anti-Inflammatory Medications: Since inflammation plays a significant role in vascular dysfunction, nonsteroidal anti-inflammatory drugs (NSAIDs) and other anti-inflammatory medications may be used to reduce inflammation and alleviate vascular damage [74,79]. Indeed, aspirin has long been used as a preventive measure for cardiovascular diseases due to its anti-inflammatory and anticoagulative effects [115].Lifestyle Modifications: Lifestyle changes, such as adopting a heart-healthy diet, increasing physical activity, and quitting smoking, can improve vascular function and reduce the risk of degenerative diseases [116].Exercise Programs: Regular exercise has a positive impact on vascular health. Exercise can improve endothelial function, increase nitric oxide production, and enhance blood vessel flexibility [116].Cholesterol Management: High levels of cholesterol, especially low-density lipoprotein (LDL) cholesterol, can contribute to atherosclerosis and vascular dysfunction [33]. Statins and other lipid-lowering medications are used to manage cholesterol levels [117].Antiplatelet Therapy: Antiplatelet medications, such as aspirin, can help prevent blood clots and reduce the risk of vascular events like heart attacks and strokes [118,119].Stem cell therapy: In recent years, stem cell therapy has emerged as a potential treatment option for degenerative diseases of the future. Stem cells, including embryonic stem cells, induced pluripotent cells, and tissue-specific stem cells, possess remarkable regenerative properties. These properties enable them to repair damaged tissues caused by degenerative diseases and restore the full functionality of organs and tissues. Additionally, numerous stem cell types exhibit immunomodulatory, angiogenic, and cell-protective properties, which could contribute to the restoration of blood vessel integrity and tissue health [120,121].

It is important to note that the choice of therapy depends on the specific degenerative disease, its stage, and individual patient factors. Treatment plans are typically developed in consultation with healthcare professionals who can tailor interventions to the patient’s needs. Additionally, a holistic approach that combines multiple strategies, including lifestyle modifications and medication, is often the most effective way to address vascular dysfunction and mitigate disease progression.

## 8. Future Directions and Research Opportunities

There are several gaps in our current understanding of the complex relationship between the vascular system and degenerative diseases, highlighting areas where further research is needed:Mechanisms of Vascular Dysfunction: While we know that vascular dysfunction is a common feature in many degenerative diseases, the precise mechanisms underlying this dysfunction remain incompletely understood. Elucidating the specific molecular and cellular pathways involved in vascular damage and their crosstalk with degenerative disease pathology is crucial for developing targeted therapies.Biomarkers for Vascular Dysfunction: Identifying reliable biomarkers that can detect early signs of vascular dysfunction and degenerative diseases is an ongoing challenge. Research efforts are needed to discover and validate biomarkers that can predict the risk of degenerative diseases and monitor vascular health.Interplay with Genetics: The influence of genetic factors on vascular dysfunction in degenerative diseases is not fully understood. Further research is required to unravel the genetic predispositions that may make some individuals more susceptible to vascular complications.Role of the Microbiome: Emerging evidence suggests that the gut microbiome may play a role in many degenerative diseases [122]. Investigating the interactions between gut microbiota and the vascular system in the context of a degenerative disease could provide insights into novel therapeutic approaches.Age-Related Changes: Aging is a significant risk factor for both degenerative diseases and vascular dysfunction [16,17]. Research is needed to understand how age-related changes in vessels and tissues interact with each other in the context of structural and functional alterations that contribute to disease progression.Plasticity and Repair: A comprehensive understanding of the natural mechanisms for repair and plasticity in the context of vessels and specific tissue is essential for a deeper understanding of vascular dysfunction and degenerative disease [109]. Such an understanding is pivotal for gaining deeper insights into vascular dysfunction and degenerative diseases. Additional research is imperative in this field.Impact of Environmental Factors: The role of environmental factors, such as pollution, diet, lifestyle, and climate change, in vascular dysfunction and degenerative diseases needs further exploration [123]. Understanding how these factors interact with genetic and biological mechanisms is vital.Long-Term Outcomes: Many studies focus on short-term outcomes of vascular interventions. There is a need for long-term follow-up to assess the durability and sustained benefits of vascular-focused therapies in degenerative diseases.Integration of Data: Combining data from various sources, such as genomics, proteomics, and clinical records, can provide a comprehensive view of vascular dysfunction in degenerative diseases. Advanced data integration techniques and artificial intelligence can help uncover hidden patterns and relationships.Ethnic and Racial Disparities: Investigating ethnic and racial disparities in the prevalence and impact of degenerative diseases and vascular dysfunction is essential for addressing health inequities and tailoring treatments to diverse populations.Translational Research: Bridging the gap between basic science research and clinical application is crucial. Further efforts are needed to translate promising laboratory findings into effective treatments for patients with degenerative diseases.

In all, the complex relationship between the vascular system and degenerative diseases is an area of ongoing research. Addressing these gaps in understanding will pave the way for innovative strategies to prevent, diagnose, and treat these diseases, ultimately improving the quality of life for affected individuals.

## 9. Concluding Remarks

Thus, it is evident that the vascular system is not merely a passive conduit for blood flow but an active player in the pathogenesis and progression of various degenerative conditions affecting multiple organ systems. Researchers and clinicians in the field of degenerative disease should consider the pivotal role of vascular health as a potential therapeutic target and a key player in disease prevention. The insights gained from this review underscore the urgent need for multidisciplinary collaboration and innovative approaches to tackle degenerative diseases comprehensively. Moreover, the integration of vascular assessments and interventions into clinical practice becomes imperative. We hope that this review will encourage continued exploration and further research in the field and guide the development of precision medicine strategies that consider the vascular dimension in the management of degenerative diseases.

## Figures and Tables

**Figure 1 ijms-25-02169-f001:**
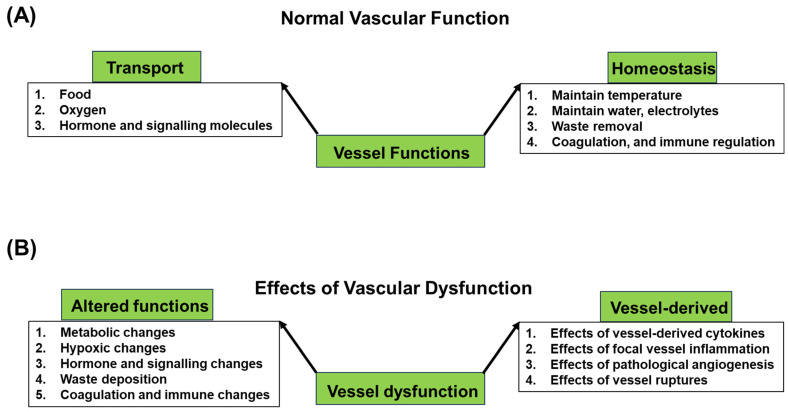
Normal vessel functions and effects of vascular dysfunctions on degenerative diseases. (**A**) The main functions of blood vessels encompass the transportation of oxygen, food and nutrients, and biomolecules, including hormones, to tissues and organs. Another crucial role is maintaining organ and tissue homeostasis for proper functionality. Functioning tissues and organs produce waste products, which are primarily cleared through vessels. Additionally, vessels play a vital role in tightly regulating temperature and electrolyte levels. Additionally, the vascular endothelium exhibits vasodilatory, anti-proliferative, anti-coagulative, and anti-inflammatory properties. Under appropriate signals, it can become proliferative for angiogenesis, contribute to inflammation through expression of adhesion molecules and cytokines, and aid in coagulation, which are essential for healing processes. (**B**) Dysfunctional vessels, by hindering the transportation of food, electrolytes, and nutrients, can induce metabolic changes in organs and tissues. Vascular dysfunction may lead to oxygen deprivation, causing hypoxic changes in tissues and organs. Failure to transport hormones can also impact organ functions. The anti-inflammatory and anticoagulative properties may alter vascular dysfunction in a way that influences degenerative diseases. Importantly, the failure to clear waste through vessel processes can trigger degenerative diseases, particularly neurodegenerative diseases. Local vascular inflammation or vessel rupture can also impact tissues and organs, potentially resulting in degenerative diseases. Notably, pathological angiogenesis can initiate several degenerative processes.

**Figure 2 ijms-25-02169-f002:**
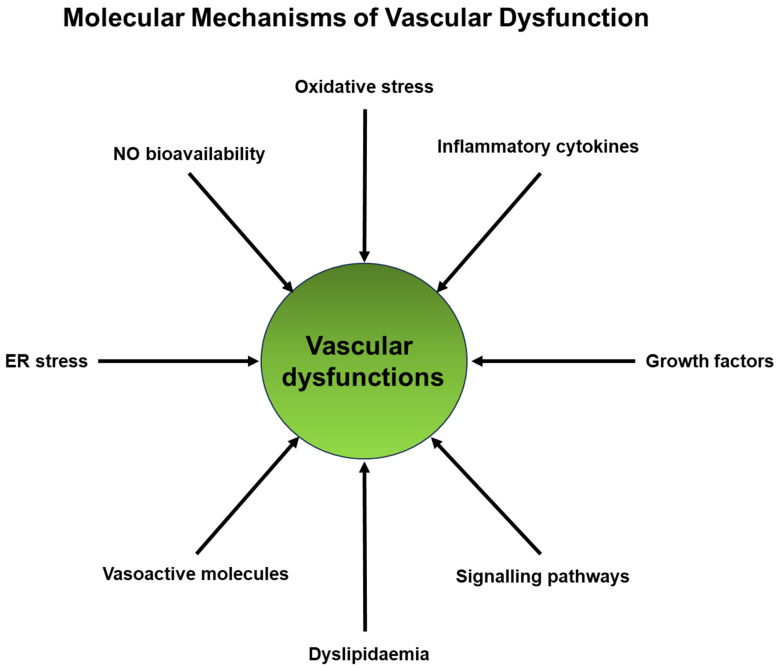
Molecular mechanisms of vascular dysfunction. Several mechanisms of vascular dysfunction have been implicated, and these mechanisms play a crucial role in vascular conditions such as atherosclerosis and hypertension. One significant molecular aspect of vascular dysfunction is endothelial dysfunction. The causes of endothelial dysfunction include reduced nitric oxide (NO) bioavailability, vascular inflammation, and dyslipidemia. In the context of vascular inflammation, endothelial cells can express adhesion molecules and cytokines, impacting the degenerative processes of organs and tissues. Additionally, growth factors have the potential to induce smooth muscle cell proliferation and extracellular matrix production, leading to sclerosis and hypoperfusion. Vasoactive molecules such as endothelin-1 or Angiotensin II can also exert effects on vessels. Moreover, certain signaling systems have been identified as crucial contributors to vascular dysfunction.

**Figure 3 ijms-25-02169-f003:**
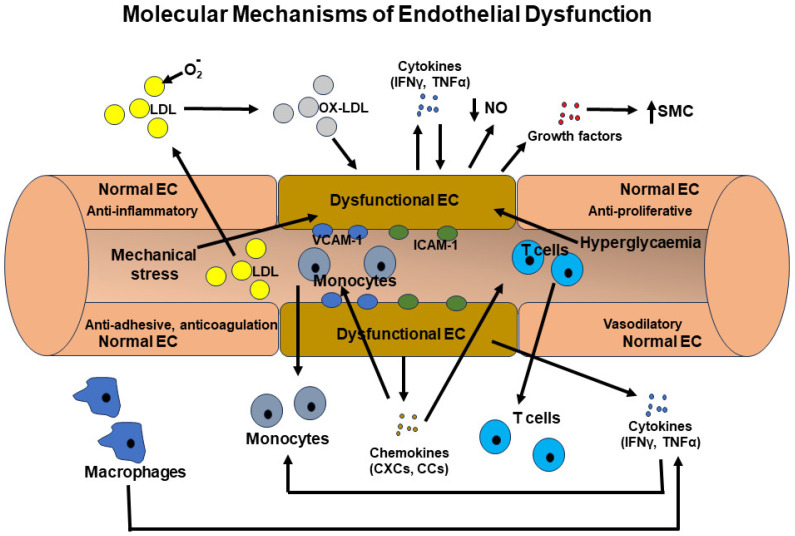
Molecular mechanisms of endothelial dysfunction. In a healthy state, endothelial cells exhibit anti-proliferative, vasodilatory, anti-inflammatory, anti-coagulant, and anti-inflammatory cell adhesion properties. However, several conditions, including elevated levels of low-density lipoprotein (LDL), mechanical stress on the vessel wall caused by hypertension or turbulent flow at arterial bifurcations, chronic generalized inflammatory conditions, hyperglycemia, and related products, can trigger endothelial dysfunction. Elevated LDL levels in the bloodstream can infiltrate the vessel wall where they become oxidized. Oxidized LDL activates endothelial cells to express cytokines (INFγ, TNFα), chemokines (MCP-1, IP-10), adhesion molecules (ICAM-1, VCAM-1), and growth factors (PDGF), which act in an autocrine and paracrine manner. Additionally, oxidized LDL acts as a chemoattractant for inflammatory macrophages. The chemokines produced by endothelial cells, along with oxidized LDL, induce the chemotactic accumulation of inflammatory cells, including macrophages and T cells. Furthermore, cytokines, in conjunction with oxidized LDL, activate macrophages and T cells to produce more cytokines and chemokines. Activated endothelial cells and inflammatory cells also produce growth factors that stimulate the proliferation of vascular smooth muscle cells and remodel the vessel wall. These smooth muscle cells are also activated by cytokines and participate in the inflammatory process. Additionally, inflammatory cells produce various proteases that remodel the vessel wall and affect the local inflammatory condition and coagulation system. Moreover, cytokines and oxidized LDL inhibit the production of nitric oxide (NO) from endothelial cells, which is one of the main vasodilatory signals for vessels. These events initiate a vicious cycle whereby endothelial cells acquire proliferative, inflammatory cell adhesive, pro-coagulant, reduced vasodilation, and inflammatory cytokine, chemokine, and growth factor-producing properties. ICAM-1 = intercellular adhesion molecule-1, VCAM-1 = vascular cell adhesion molecule-1, INFγ = interferon γ, TNFα = tumor necrosis factor α, MCP-1 = monocyte chemoattractant protein-1, IP-10 = INFγ-induced protein-10, PDGF = platelet-derived growth factor.

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
