# Peer review of "The Role of the Vascular System in Degenerative Diseases: Mechanisms and Implications"

_ijms, 2024, doi:10.3390/ijms25042169_

Round 1
Reviewer 1 Report
Comments and Suggestions for Authors
The review provides a concise summary of the pathophysiology involving the vascular system and its risk factors, making it a valuable resource for beginners in the field. However, reviews focusing solely on such pathologies have been reported extensively in the past.
Additionally, this review lacks specific numerical data, somewhat diminishing its impact in emphasizing the importance of the vascular system to readers.
To enhance the originality of this review, it would be beneficial to incorporate recent statistics related to:
*The global impact of cardiovascular diseases in terms of mortality and, the economic losses incurred.
*Current prevalence and potential forecasts of cardiovascular diseases worldwide.
By integrating such information into the introduction, the review would gain a unique perspective, making a stronger case for the significance of the vascular system. Additionally, the lack of uniform indentation gives the impression that it is unorganized, and it significantly lacks readability as a review article. It would be beneficial to improve this point. It would be better to align each item with the indentation that appears on pages 6-15.
There were no sentences that were extremely difficult to read, but it would be a good idea to proofread the English text just to be sure.
Author Response
Reviewer 1
The review provides a concise summary of the pathophysiology involving the vascular system and its risk factors, making it a valuable resource for beginners in the field. However, reviews focusing solely on such pathologies have been reported extensively in the past. Additionally, this review lacks specific numerical data, somewhat diminishing its impact in emphasizing the importance of the vascular system to readers. To enhance the originality of this review, it would be beneficial to incorporate recent statistics related to:
- The global impact of cardiovascular diseases in terms of mortality and, the economic losses incurred.
Response: We agree with the reviewer that information on cardiovascular disease in terms of mortality would be valuable for the paper. Hence, we have discussed this point in the revised version of the review paper. (page 4, line 181 to page 5, line 207)
- Current prevalence and potential forecasts of cardiovascular diseases worldwide.
Response: We agree with the reviewer that information on current prevalence and potential forecasts on cardiovascular disease would be valuable for the paper. Hence, we have discussed this point in the revised version of the review paper. (page 4, line 181 to page 5, line 207)
By integrating such information into the introduction, the review would gain a unique perspective, making a stronger case for the significance of the vascular system.
- Additionally, the lack of uniform indentation gives the impression that it is unorganized, and it significantly lacks readability as a review article. It would be beneficial to improve this point. It would be better to align each item with the indentation that appears on pages 6-15.
Response: According to the reviewer’s suggestion, we have reformatted the revised manuscript.
Reviewer 2 Report
Comments and Suggestions for Authors
Congratulations for the excellent article, this article makes the connection between the Vascular System and Degenerative Diseases, in a very clear way and tries to go into the mechanistic details. I advise you to publish this article, as it has the potential to be widely cited.
I just think you could improve the figures, as they are too simple: for example, in figure 2, put the various items of oxidative stress, signaling pathways (which are the main ones), etc.
In item 4, it's missing a bit about the environment and endocrine disruptors, where this link has also been demonstrated (see for example these articles( Arch Toxicol. 2024 Jan;98(1):1-73. doi: 10.1007/s00204-023-03614-0. Epub 2023 Oct 19; Int J Med Sci
. 2015 Oct 30;12(12):926-36. doi: 10.7150/ijms.13267. eCollection 2015.) although this item is only addressed in the conclusions, and like the future ones. I think it could be included in the body of the text, as there are many authors who have already addressed this subject of EDCs at the cardiovascular level and, as mentioned above.
Author Response
Reviewer 2
Congratulations for the excellent article, this article makes the connection between the Vascular System and Degenerative Diseases, in a very clear way and tries to go into the mechanistic details. I advise you to publish this article, as it has the potential to be widely cited.
- I just think you could improve the figures, as they are too simple: for example, in figure 2, put the various items of oxidative stress, signaling pathways (which are the main ones), etc.
Response: We also think that figure 2 is very simple. Hence, the detail mechanisms of endothelial dysfunction have been added in the revised manuscript (page 10)
- In item 4, it's missing a bit about the environment and endocrine disruptors, where this link has also been demonstrated (see for example these articles( Arch Toxicol. 2024 Jan;98(1):1-73. doi: 10.1007/s00204-023-03614-0. Epub 2023 Oct 19; Int J Med Sci. 2015 Oct 30;12(12):926-36. doi: 10.7150/ijms.13267. eCollection 2015.) although this item is only addressed in the conclusions, and like the future ones. I think it could be included in the body of the text, as there are many authors who have already addressed this subject of EDCs at the cardiovascular level and, as mentioned above.
Response: According to the reviewer’s suggestion, we have added the topic of the environment and endocrine disruptors in degenerative diseases have been discussed in the revised manuscript (page 12, line 509-535).